# Culinary Habits and Health: Analyzing the Impact of Cooking Practices and Knowledge Among Spanish Young Adults

**DOI:** 10.3390/nu17101635

**Published:** 2025-05-10

**Authors:** Elena Sandri, Michela Capoferri, Michela Piredda, Valentina Micheluzzi

**Affiliations:** 1Faculty of Medicine and Health Sciences, Catholic University of Valencia San Vicente Mártir, c/Quevedo, 2, 46001 Valencia, Spain; elena.sandri@ucv.es; 2Department of Animal Production and Health, Veterinary Public Health, and Food Science and Technology, Faculty of Veterinary Medicine, Institute of Biomedical Sciences, Cardenal Herrera-CEU University, Calle Santiago Ramón y Cajal 20, 46115 Alfara del Patriarca, Spain; michela.capoferricapoferri@alumnos.uchceu.es; 3Research Unit of Nursing Science, Department of Medicine and Surgery, Campus Bio-Medico di Roma University, Via Alvaro del Portillo, 21, 00128 Rome, Italy; 4Clinical and Interventional Cardiology, Sassari University Hospital, 07100 Sassari, Italy; valentina.micheluzzi@gmail.com; 5Department of Biomedicine and Prevention, University of Rome Tor Vergata, Via Montpellier, 1, 00133 Rome, Italy

**Keywords:** healthy habits, eating behaviors, food preferences, diet, food and nutrition, dietary patterns

## Abstract

***Background***: Culinary habits and knowledge play a key role in shaping dietary behaviors and overall health. In Spain, societal changes have led to a decline in traditional Mediterranean practices and, in some cases, to unhealthy lifestyles. This study explores the relationship between culinary habits, knowledge, and health behaviors among adults in Spain. ***Methods***: A cross-sectional study was conducted with 1534 participants using validated questionnaires to assess culinary habits, knowledge, and lifestyle factors. Data were collected online via snowball sampling and analyzed using the Kruskal–Wallis test and a Gaussian graphical model to explore variable associations. ***Results***: Participants with healthy culinary habits reported higher self-perceived health and greater fruit and vegetable consumption, while those with unhealthy habits consumed more fast food. Higher culinary knowledge was linked to better dietary choices, more vegetables and cereals, and lower smoking rates. Higher home cooking frequency was moderately correlated with the use of healthy cooking techniques. The network analysis showed that the frequency of cooking at home is positively associated with gender and age. A similar association was found for time spent cooking. Positive associations were also found between living with the family and time spent cooking, while age showed a negative correlation with living situation. Centrality measures identified influential variables within the network. ***Conclusions***: Culinary knowledge and habits strongly influence health behaviors. Network analysis underscores the importance of specific variables such as cooking frequency and living situation in shaping lifestyle patterns. Educational programs aimed at enhancing culinary skills could promote healthier eating behaviors and mitigate public health risks.

## 1. Introduction

Dietary patterns significantly impact overall health and well-being [1,2]. Evaluating eating behaviors and lifestyle factors such as physical activity, social habits, and rest is fundamental in nutritional epidemiology. This understanding aids in health planning, the formulation of dietary guidelines, and the implementation of targeted health interventions [3].

Dietary choices are influenced by culinary knowledge and habits [4]. Maintaining a well-balanced diet and good health requires an understanding of nutrition and the adoption of healthy culinary practices [5]. A combination of diverse and nutritious food, appropriate cooking methods, and cultural appreciation for quality meals contributes to disease prevention and overall well-being [1]. Encouraging these practices from an early age can have long-term benefits for population health.

In addition to cultural factors, dietary choices are profoundly influenced by social and emotional factors [6]. Social elements such as family dynamics, shifts in gender roles, and the increasing participation of women in the labor market can shape food preferences and meal preparation practices. Furthermore, emotional factors, such as stress, anxiety, and boredom, significantly influence food choices, often leading to unhealthy eating behaviors [6]. These emotional triggers, combined with social and cultural influences, shape eating habits. Understanding these factors is crucial for promoting healthier food choices and creating effective interventions.

Spain has a rich gastronomic tradition, yet recent studies suggest a shift in dietary patterns [7,8]. Although the Mediterranean diet remains prevalent [9], its traditional elements are increasingly being replaced by other dietary trends. Recognized as one of the healthiest dietary models [10], the Mediterranean diet emphasizes high consumption of fruits, vegetables, legumes, nuts, olive oil, and fish, with limited intake of red meat and refined sugars. This diet provides essential nutrients such as vitamins, minerals, and antioxidants, contributing to its anti-inflammatory and antioxidant properties [11,12,13,14]. Regular adherence to the Mediterranean diet is associated with a lower incidence of metabolic and vascular diseases and supports cognitive health through nutrients like omega-3 fatty acids [15,16].

Beyond food selection, culinary habits and cooking techniques play a crucial role in determining a meal’s health benefits [17]. Culinary competencies encompass not only technical skills but also knowledge of food properties, menu planning, and resource management [18,19]. These skills reflect confidence, attitudes, and the ability to apply knowledge effectively in food preparation.

Although culinary skills have traditionally been considered domestic knowledge, recent societal changes have significantly influenced cooking habits [20]. Increased work demands and a higher proportion of women in the workforce have contributed to a rise in fast food consumption, pre-packaged meals, and a decline in home-cooked meals [21,22,23]. Additionally, meal schedules have been adjusted to accommodate modern work patterns, impacting food choices and cooking frequency.

Time constraints and stress levels have led many individuals to prioritize convenience foods over healthier options requiring longer preparation [24,25,26]. Promoting culinary education and home-cooked meal preparation is essential, as these practices are associated with better dietary quality and improved health outcomes [17]. Healthy cooking methods, such as steaming, roasting, and grilling, help preserve nutrients while reducing the intake of unhealthy fats. However, it is important to note that certain cooking techniques, like grilling at high temperatures, can lead to the formation of potentially harmful compounds such as heterocyclic amines (HCAs) and polycyclic aromatic hydrocarbons (PAHs), which have been associated with increased cancer risk [27]. Studies suggest that lower culinary skills are linked to overweight and obesity [28], while limited cooking abilities among older adults correlate with unhealthy dietary behaviors and nutritional deficiencies [29]. Additionally, increased time spent cooking is associated with healthier eating habits [30].

Nutritional education programs aimed at improving eating behaviors and culinary skills should be systematically designed and theory-based to maximize effectiveness [31]. According to the Precede–Proceed model, interventions should begin with an assessment of social and environmental factors influencing health (classified into predisposing, enabling, and reinforcing factors), followed by targeted educational and policy initiatives [31]. This framework emphasizes the dynamic interactions between personal, behavioral, and environmental factors, highlighting the need for structured planning in nutrition education programs [32].

### Study Aim

This study aims to examine the culinary knowledge and skills of young Spanish adults and their relationship to nutritional and lifestyle habits.

## 2. Materials and Methods

### 2.1. Study Design and Sample Size Calculation

This cross-sectional study design utilized data from the Continuous Population Statistics (Estadística Continua de Población, ECP) provided by the Spanish Statistics Institute [33] to determine the sample size. Based on Spain’s population as of 1 January 2023 (41,995,741 individuals), the minimum required sample size was calculated using a 95% confidence interval, a 5% margin of error, and a maximum variability assumption (*p* = *q* = 50%), resulting in 385 surveys. To enhance the study’s robustness, the final sample size was expanded to nearly four times the minimum requirement. This study was conducted in accordance with the STROBE (Strengthening the Reporting of Observational Studies in Epidemiology) guidelines and the Extension for Nutritional Epidemiology (STROBE-nut) to ensure comprehensive and transparent reporting of the methodological and nutritional aspects [34,35].

### 2.2. Eligibility Criteria

Inclusion criteria included the following:-Individuals aged 18 years or older;-Spanish citizens residing in Spain.

Exclusion criteria included the following:
-Individuals with chronic conditions that affect their dietary habits (such as diabetes, celiac disease, inflammatory bowel diseases, and severe food allergies or intolerances);-Participants experiencing temporary disruptions in their diet (e.g., hospitalization, incarceration, etc.);-Individuals unable to cook due to physical or mental limitations.

### 2.3. Ethical Considerations

The study adhered to the Declaration of Helsinki [36] and received approval from the Research Ethics Committee of the Catholic University of Valencia (approval code UCV/2023-2024/192, 28 May 2024). Prior to participating, explicit informed consent was obtained from all individuals, ensuring they were fully informed about the study’s objectives, procedures, and potential risks, emphasizing the voluntary nature of their participation.

### 2.4. Data Collection

The questionnaire was hosted on Google Forms and distributed through snowball sampling [37], primarily through Instagram (@elretonutricional), alongside LinkedIn, Twitter, WhatsApp, and student networks. Data collection occurred from June 2024 to November 2024.

### 2.5. Instruments

Two validated questionnaires were used.

The instrument used to collect information on the culinary knowledge of the population was the questionnaire translated and culturally adapted by researchers at the University of Blanquerna [38], based on the original in English, developed and validated in Ontario [39].

The 23-item NutSo-HH questionnaire (Nutritional and Social Healthy Habits) was used to assess nutritional and lifestyle habits. [40] The NutSo-HH questionnaire consists of 23 items that collect data on the nutritional, health, and lifestyle habits, including dietary choices and consumption frequency. It is a multidimensional instrument encompassing six primary factors: F1 (Mediterranean foods), F2 (healthy and unhealthy foods), F3 (meats and dairy products), F4 (eating disorders), F5 (rest habits), and F6 (alcohol consumption). Additionally, two overarching categories are included: NUTRI (a combination of F2 and F3) and HH (a combination of F4 and F5).

To minimize data loss due to human error during questionnaire completion, most questions were designed as closed-ended, multiple-choice items, allowing respondents to select the option that best reflected their lifestyle habits or frequency of food consumption.

Socio-demographic data were also collected. Finally, culinary habits were examined through nine questions, including sources of cooking inspiration and self-taught skills.

Nine qualitative questions, developed with expert input, were added to explore participants’ culinary habits and preferred cuisine types. The content validity of these additional questions was assessed by a panel of eleven experts in nutrition and health promotion, comprising three nurses, three nutritionists, a psychologist, three university professors in health sciences, and a chef. To ensure face validity and readability, the questionnaire was pilot-tested on a sample of the target population (*n* = 43). Structured cognitive interviews were conducted to evaluate completion time, clarity, readability, comprehensiveness, acceptability, and the questionnaire’s formal aspects.

### 2.6. Measurements

#### 2.6.1. Socio-Demographic Variables

The study collected several socio-demographic variables, including sex (analyzed in binary form: male or female), age (categorized as young (18–30 years), adults (31–65 years), and seniors (>65 years)), and educational level (classified as basic education (no formal education, primary or secondary education, vocational training, or baccalaureate) and higher education (bachelor’s degree, master’s degree, or Ph.D.)). Income level was grouped into three categories: low income (≤EUR 2200/month), medium-high income (>EUR 2200/month), and not answered. Following the criteria used in other articles [41,42] where it has been found that there is a difference in the nutritional pattern of the population depending on the size of the municipality of residence, municipality size was classified into three groups: populations of <2000 inhabitants, 2000–10,000 inhabitants, and >10,000 inhabitants. Living arrangements were classified as living alone or living with others, while family living was further categorized as residing in the family home or living independently.

#### 2.6.2. Culinary Habits

Culinary habits were assessed using nine questions. Two qualitative questions explored where respondents consulted cooking recipes and where they learned how to cook. The remaining seven quantitative questions were categorized according to the criteria shown in Table 1.

The classification of cooking techniques as “healthy” or “unhealthy” is based on their impact on the nutritional quality of food, considering factors such as nutrient retention, the formation of harmful compounds, and their effects on health. Cooking methods that help preserve nutrients, limit the use of unhealthy fats, and minimize the production of harmful substances are considered beneficial. These include steaming, boiling (with proper control of time and water usage), stewing, braising, and roasting or grilling at moderate temperatures. Conversely, cooking techniques that reduce nutrient content, generate harmful compounds, or increase the intake of unhealthy fats are classified as detrimental. Examples include deep frying, roasting or grilling at high temperatures, and using saturated or trans fats in cooking [43,44,45,46]. The term “convenience foods” refers to pre-cooked, canned, or preserved products that require the addition of additives and preservatives during processing, which can alter their nutritional value and impact on health. The first four questions explore cooking habits beneficial to health, while questions 5, 6, and 9 describe habits with a negative impact. The final culinary habit score is calculated by summing the scores from the first four questions and subtracting those from questions 5, 6, and 9. The total possible score ranges from −10 to 17 points, spanning a 27-point range. Based on their score, participants were classified into three categories: healthy culinary habits (HCH): 9 to 17 points; needs improvement (NI): −1 to 8 points; and unhealthy culinary habits (UCH): −10 to −2 points.

#### 2.6.3. Culinary Knowledge

Participants’ confidence in culinary skills was measured using a 5-point Likert scale, ranging from 1 (“not very confident”) to 5 (“very confident”). The total culinary knowledge score ranged from 0 to 90 points. Participants were classified as follows: poor culinary knowledge (PCK): ≤30 points; average culinary knowledge (ACK): 31–60 points; good culinary knowledge (GCK): >61 points.

#### 2.6.4. Potential Eating Disorders Variables

Potential indicators of eating disorders were assessed through questions related to worry about weight gain, feeling overweight, difficulty controlling food intake, feeling of shame after eating, and body image concerns. These variables were measured using a 6-point Likert scale ranging from 1 to 6, where 6 indicated “always”, 5 “very frequently”, 4 “frequently”, 3 “occasionally”, 2 “rarely”, and 1 “never”. Additionally, participants were asked whether they had ever received a formal diagnosis of an eating disorder.

#### 2.6.5. Nutritional, Health, and Lifestyle Variables

Following established methodologies from previous studies [47,48], nutritional and lifestyle variables were evaluated using a 4-point Likert scale ranging from 1 (the lowest frequency of consumption or habit) to 4 (the highest frequency of consumption or habit). These variables were analyzed without specific units of measurement. Body mass index (BMI) was calculated as weight (kg) divided by height squared (m^2^). Physical activity was measured in minutes of sport or exercise per week.

### 2.7. Data Analysis

The collected data were entered into a Microsoft Excel database, where they were carefully reviewed to correct any errors and inconsistencies, particularly focusing on entry mistakes and outliers. Some variables were categorized or derived from others. Extreme BMI values (below 14 and above 40) were excluded to ensure data reliability. Once processed, the data were transferred to Jamovi Version 2.3.28.0 [49] for statistical analysis. The normality of variables was assessed using the Shapiro–Wilk test, and Q-Q plots confirmed that none of the variables met normality assumptions [39].

Consequently, non-parametric tests were applied: chi-square test for categorical variables, and the Kruskal–Wallis test for independent ordinal or numeric variables. The significance level was set at *p* < 0.05. For statistically significant comparisons, Dunn’s sequentially rejective Bonferroni correction was applied as a post hoc test to determine which groups differed significantly.

Spearman’s correlations were calculated to examine relationships between culinary habits and culinary knowledge, highlighting statistically significant correlations.

Finally, to explore how dietary behavior interacts with various socio-demographic factors, a network analysis was employed. In this model, each node (represented as a circle) corresponds to a specific variable, and the edges (lines linking the nodes) depict the strength and direction of their associations. The visual characteristics of these edges—such as their thickness and color—convey the intensity and polarity of the relationships.

Spearman correlation coefficients were used to assign weights to the edges. To ensure that only direct associations were retained and to reduce the influence of indirect or misleading correlations, the analysis was conducted using a Gaussian graphical model (GGM), selected through the extended Bayesian information criterion (EBIC) [50]. This technique enables the estimation of partial correlations by adjusting for the effect of other variables, thereby enhancing the precision of the resulting network.

The use of the EBIC also introduces a regularization mechanism, promoting a simpler and more stable network structure and reducing the risk of overfitting. This analytical framework makes it easier to identify meaningful relationships within the data, supporting clearer interpretation of complex interdependencies by both researchers and lay audiences.

## 3. Results

### 3.1. Socio-Demographic Characteristics

Table 2 presents the socio-demographic characteristics of the sample. Approximately one-third of the participants were men, while two-thirds were women. The mean age of participants was 39.9 years, with women slightly older than men (40.6 vs. 38.6 years). Youn adults (18–30 years) accounted for 36.4% of the sample, adults (31–65 years) made up 61.3%, and only 2.3% were aged over 65. Regarding education, 41.9% of the sample had a basic education, while 58.1% had higher education qualifications. Most of the participants lived with others (88.9%), most commonly with family members (78.9%).

### 3.2. Cooking Habits

Most participants self-reported cooking at home every day or almost every day (Table 3). On average, they reported spending approximately 1.5 h per day on cooking. Participants also reported frequently using household appliances for healthy cooking techniques (mean score: 3.08 out of 5) and engaging in healthy cooking practices always or very often (mean score: 3.84 out of 5). Furthermore, they self-reported limiting the use of unhealthy sauces and condiments (mean score: 1.93 out of 5) as well as pre-packaged or canned foods (mean score: 1.56 out of 5).

### 3.3. Culinary Knowledge

Participants generally self-reported a high level of culinary knowledge, with individual item scores ranging from 3.5 to 4.13 on a 5-point scale. Reported strengths included knowledge related to cooking specific food groups such as vegetables (mean ± SD: 4.13 ± 1.13), cereals (4.10 ± 1.13), and proteins (4.10 ± 1.10) as well as the use of kitchen equipment (4.09 ± 1.12). In contrast, tasks such as weekly menu planning (3.22 ± 1.39), reading nutritional information (3.57 ± 1.36), interpreting recipes (3.59 ± 1.35), and using leftovers (3.68 ± 1.33) appear to be reported as slightly more challenging. The overall self-reported culinary knowledge score was 61.2 ± 17.0 on a scale ranging from 0 to 90 (see Table 4 and Figure 1).

### 3.4. Health Variables and Nutritional and Lifestyle Habits of Sample

Table 5 summarizes the self-reported scores related to health, nutritional habits, and lifestyle variables in the sample. Participants had a mean BMI of 24.2 ± 4.26, consistent with a normal weight range, and a relatively high level of self-perceived health (mean score: 3.98 out of 5). Self-reports regarding potential indicators of disordered eating showed average scores around 3, suggesting that experiences such as fear of gaining weight, loss of control over food intake, or dissatisfaction with body image may occur only occasionally.

The reported weekly minutes of physical activity—both vigorous (96.7 ± 208) and moderate (118 ± 337)—appear to meet WHO recommendations, although responses showed considerable variability. Alcohol and tobacco use were generally reported as low, with most participants indicating alcohol consumption less than once per week and little to no smoking.

Participants reported sleeping between 6 and 7.5 h per night on average, with sleep quality scores ranging between 3 and 4 out of 5, indicating that they often feel rested upon waking.

In terms of nutritional habits, the reported frequency of fruit consumption (mean: 2.35 ± 0.75) corresponded to approximately one to four pieces per day, while vegetable intake (3.45 ± 0.75) ranged from two portions per week to one portion per day. The reported consumption of white fish (1.72 ± 0.57) and oily fish (1.84 ± 0.59) was relatively low, generally between never and twice a week. White meat (2.54 ± 0.68) was consumed more frequently than red meat (1.74 ± 0.69).

The intake of unhealthy foods, such as fast food (2.43 ± 0.76), fried foods (2.32 ± 0.78), and ultra-processed products (2.31 ± 0.91), was reportedly infrequent, typically 2–4 times per month. Similarly, sugary drinks (1.43 ± 0.66), fruit juices (1.23 ± 0.54), and energy drinks (1.06 ± 0.30) were consumed infrequently. In contrast, coffee consumption (1.76 ± 0.71) corresponded to approximately two cups per day.

### 3.5. Health Variables and Nutritional and Lifestyle Habits Compared to Cooking Habits

The comparison of health variables and nutritional and lifestyle habits across cooking habit categories (unhealthy culinary habits—UCH, need improvement—NI, and healthy culinary habits—HCH) revealed significant differences in several parameters.

Body mass index (BMI) showed a statistically significant difference among groups (*p* = 0.043), with the UCH group presenting the lowest mean BMI (23.3 ± 4.85) compared to NI (24.3 ± 4.19) and HCH (24.0 ± 4.34). Self-perceived health was higher (*p* < 0.001) in the HCH group (4.07 ± 0.70) compared to the NI (3.89 ± 0.78) and UCH groups (3.87 ± 0.99). Obesophobia scores were highest in the HCH group (3.08 ± 0.70) and lowest in the UCH group (2.27 ± 1.33; *p* = 0.017).

The consumption of fruit and vegetables was significantly higher (*p* < 0.001) in the HCH group (fruit: 2.46 ± 0.73; vegetables: 3.65 ± 0.60) compared to the UCH group (fruit: 1.93 ± 0.80; vegetables: 3.07 ± 0.70). Conversely, the UCH group showed higher consumption of fast food (3.00 ± 0.93), fried food (3.40 ± 0.74), and ultra-processed food (3.07 ± 0.96) compared to the HCH group (fast food: 2.29 ± 0.72; fried food: 2.08 ± 0.69; ultra-processed food: 2.15 ± 0.85), all with *p* < 0.001.

Differences were observed in alcohol consumption and smoking habits, with the HCH group reporting lower alcohol consumption (1.74 ± 0.82) and smoking scores (1.22 ± 0.60) compared to the NI and UCH groups (*p* = 0.003 and *p* < 0.001, respectively). The HCH group also reported significantly higher sleep quality (3.39 ± 0.79) compared to the NI group (3.24 ± 0.89; *p* = 0.003).

Finally, participants in the HCH group reported significantly lower consumption of sugary drinks (1.31 ± 0.59) and energy drinks (1.04 ± 0.22) compared to the UCH and NI groups (*p* < 0.001 and *p* = 0.007, respectively).

### 3.6. Correlation Matrix Between the Variables of Culinary Habits and Nutritional and Lifestyle Habits

To study in more detail the relationship between the variables, a correlation matrix (Figure 2) was made between the nutritional habits and lifestyle variables that presented statistically significant differences in Table 6 and the cooking habits variables. For the sake of clarity, only the relationships that presented a certain degree of correlation are reported, with those relationships left blank in which the correlation was <0.200.

Some relationships are found between lifestyle and culinary habits. Low correlations (0.20–0.39) were observed between alcohol consumption and getting drunk (0.360) and between sleep quality and self-perceived health (0.282). Night outings showed weak correlations with alcohol consumption (0.202) and sleep quality (0.269).

Among dietary habits, moderate correlations (0.40–0.59) appeared, such as between fried food consumption and unhealthy condiments (0.418). Low correlations were found between blue and white fish consumption (0.223) and between cereal and dairy consumption (0.207).

Home cooking frequency correlated modestly with cooking time (0.492) but negatively with unhealthy cooking techniques (−0.205). Healthy cooking techniques showed moderate correlations with cooking time (0.254) and unhealthy condiments (0.242), suggesting an interaction between culinary behaviors and preparation quality.

The strongest associations was found between frequent use of unhealthy condiments and sauces and convenience food consumption (0.421), highlighting a possible preference for processed food habits in this group.

### 3.7. Health Variables and Nutritional and Lifestyle Habits Compared to Culinary Knowledge

Table 7 highlights the relationship between culinary knowledge levels (poor, average, and good culinary knowledge) and various health, nutritional, and lifestyle variables. Participants with good culinary knowledge exhibited better self-perceived health (*p* = 0.009) and reported higher sleep quality (*p* = 0.010).

Differences were observed in dietary habits, with higher consumption of vegetables (*p* < 0.001), cereals (*p* < 0.001), and blue fish (*p* < 0.001) among respondents with good culinary knowledge. Lower intake of fast food (*p* = 0.001), fried food (*p* < 0.001), and ultra-processed food (*p* = 0.001) was noted in individuals with higher culinary knowledge.

Smoking rates were significantly lower among participants with good culinary knowledge (*p* = 0.018). Finally, energy drink consumption was reduced among those with higher culinary knowledge (*p* = 0.005).

### 3.8. Gaussian Network Analysis of the Relationships Between Culinary Habits and Socio-Demographic Variables

Figure 3 displays the Gaussian network diagram representing the relationships between culinary habits and socio-demographic variables. This diagram visualizes the correlations among the study variables, with each node corresponding to a specific variable or component identified through the analysis. The edges (lines connecting the nodes) reflect the strength and direction of the correlations between them.

Correlations are color- and thickness-coded: Green lines indicate positive correlations, while red lines denote negative correlations. The thicker the line, the stronger the association between the connected variables. Lighter shades, regardless of color, indicate correlations of lower magnitude.

This type of visualization facilitates a global overview of the interdimensional relationships within the dataset, allowing the identification of behavioral patterns, clusters of related variables, and potential compensatory effects between positive and negative components of the observed behaviors (Appendix A in Appendix A shows the centrality measures of the network analysis for the different variables of the study).

Notably, some variables occupy a more central position within the network. For example, Hcf (home cooking frequency) is a highly connected node, maintaining both positive and negative correlations with several other variables—suggesting a key role in the interaction of lifestyle and demographic factors.

A strong positive correlation can be observed between Lvs (living situation) and Fml (family living), indicating that these variables are closely related and likely measure the same underlying construct. Conversely, a negative correlation between Lvs and Ag (age) reflects a logical demographic trend: adults are more likely to live independently compared to younger individuals.

As expected, there is also a strong positive correlation between Hcf and Ckt (cooking time), suggesting that individuals who cook more frequently at home also tend to spend more time cooking.

Finally, the negative relationship between Hcf and Fml implies that individuals living with their families tend to cook less frequently—possibly because younger people living at home often rely on their parents to handle meal preparation.

## 4. Discussion

This study examined the culinary knowledge and skills of the Spanish population and their association with nutrition and lifestyle habits. The findings highlight the importance of culinary behaviors as key determinants of dietary quality and, by extension, public health. These results are particularly relevant given the increasing prevalence of metabolic diseases, such as obesity, type 2 diabetes, and cardiovascular diseases, which are strongly associated with poor nutrition and unhealthy culinary practices [51].

The demographic characteristics of the sample provide valuable insights into the societal and cultural context of cooking practices in young adult Spaniards. The sample, composed mainly of women (67%) with a mean age of 39.9 years, reflects previous studies highlighting the predominant female role in domestic food preparation responsibilities, although this trend is evolving. It also aligns with a demographic group characterized by well-established family roles and stable dietary habits [52,53]. Furthermore, individuals in this age cohort are typically more aware of health and nutrition, as they are more likely to have access to health-related information and resources. A significant portion of participants (over 50%) possess a higher level of education, suggesting that this is associated with a more informed and intentional approach to health as well as greater attention to self-care. Individuals with higher levels of education are generally more informed about dietary practices and nutritional guidelines, which may positively influence their culinary behaviors and decisions related to food preparation. However, education alone does not necessarily translate into cooking confidence or skills. Individuals with similar educational levels may exhibit varying degrees of culinary competence due to factors such as culture, socioeconomic status, or geographic location [54]. Therefore, while education can facilitate healthier choices, it may not be sufficient on its own, and complementary interventions might be required to address persistent barriers such as time constraints, limited access to fresh ingredients, or deeply rooted culinary habits.

The data also indicate a strong prevalence of home cooking among the Spanish population, with an average frequency score of 4.08 and an average cooking time of about an hour and a half per day. These habits are positively correlated with healthier dietary patterns, as evidenced by the higher consumption of fruits and vegetables among individuals with healthy culinary habits (HCH). These findings align with the literature that highlights the beneficial impact of home cooking on dietary quality, particularly in promoting the intake of nutrient-dense foods. The high frequency of using healthy cooking techniques (mean score of 3.84) further supports the notion that the sample is inclined toward healthier cooking practices. However, the reported use of unhealthy cooking techniques (mean score of 1.91) and the addition of unhealthy condiments and sauces (mean score of 1.93) indicate areas for improvement. These scores suggest that while the population demonstrates a strong tendency toward healthier cooking methods, there is still a notable reliance on less healthy practices, such as the use of unhealthy condiments or occasional consumption of ultra-processed foods, which could undermine dietary quality. Encouragingly, the limited reliance on convenience foods (mean score of 1.56) suggests an opportunity to further promote healthier home cooking practices within the population. Moreover, participants with higher culinary knowledge demonstrated better dietary habits and lifestyle outcomes, including increased consumption of vegetables and cereals, while showing a reduced reliance on fast and ultra-processed foods. These findings align with previous studies that identified strong correlations between culinary skills and healthier eating behaviors [28]. Furthermore, individuals with strong culinary knowledge exhibited lower smoking rates and better self-perceived health, suggesting that culinary competence may extend beyond food preparation to influence broader health-related decisions [29].

In addition to dietary behaviors, this study also assessed key lifestyle variables such as alcohol and tobacco consumption, which showed meaningful associations with culinary practices. Participants with healthy culinary habits (HCH) reported lower alcohol consumption (mean score of 1.74) and lower smoking rates (mean score of 1.22) compared to those with unhealthy or suboptimal culinary habits. These findings are consistent with previous research linking greater culinary competence to healthier lifestyle choices, including improved dietary patterns and self-care routines, which could potentially extend to the reduced use of harmful substances [55]. This study presents several strengths that enhance its reliability and contribution to the existing literature. The survey instrument was tailored to capture a multidimensional perspective on culinary knowledge, skills, and their association with dietary and lifestyle habits. The inclusion of variables such as education level, gender, and age enabled a nuanced analysis that provides valuable insights into the demographic and sociocultural factors influencing culinary behaviors. This research highlights the role of culinary habits in improving dietary quality, making it a valuable resource for the initial phase of the Precede–Proceed framework. It helps identify factors influencing health behaviors and demonstrates how culinary knowledge and skills contribute to better dietary habits, aligning with the framework’s focus on social, behavioral, and environmental factors affecting health outcomes [31,32].

However, some limitations of the study should also be acknowledged. One possible limitation is the lack of an in-depth analysis of regional culinary differences within Spain. Given the country’s diverse culinary traditions, future studies should aim to capture these regional variations and explore how they impact dietary quality. The cross-sectional design also limits causal inferences, as it is unclear whether culinary skills lead to healthier habits or if individuals with healthier lifestyles are more inclined to develop culinary competencies. Furthermore, the reliance on self-reported data introduces the risk of social desirability bias, particularly in assessing cooking frequency and dietary quality. Future studies could benefit from longitudinal designs to better establish causal relationships and incorporate objective measures, such as dietary records or biomarker assessments.

A limitation of this study is the gender imbalance among participants, with 67.0% being women and only 33.0% men. This disparity may have introduced a bias in the results, particularly given that this is a study on culinary practices—an area where gender roles and social expectations can influence behaviors and experiences. Although it is often observed that women participate more frequently in cooking activities, their overrepresentation may limit the generalizability of the findings to the broader population, especially to men, whose habits and perspectives may differ significantly. Future studies should aim for a more balanced sample in order to provide a more comprehensive understanding and reduce gender-related bias in data interpretation.

Another limitation is the sampling approach, which, while achieving a robust sample size, may not fully represent the broader Spanish population. In this study, the final sample is predominantly composed of adults, while the 65+ age group is underrepresented, which may limit the generalizability of the results to older populations. This underrepresentation may be due to older individuals having difficulty using electronic tools or generally having less access to social networks. Therefore, we specified in the title that the study focuses primarily on young adults.

## 5. Conclusions

This study elucidates the intricate relationships between culinary habits, culinary knowledge, and health-related behaviors within the young adult Spanish population. Individuals exhibiting healthy culinary habits (HCH) not only reported superior self-perceived health but also demonstrated significantly higher consumption of fruits and vegetables compared to those with unhealthy culinary habits (UCH). This finding supports the premise that culinary practices are determinants of dietary quality and overall health outcomes. Furthermore, the reliance on unhealthy cooking techniques and ultra-processed foods highlights critical areas for improvement, necessitating public health initiatives that promote nutritious cooking practices and reduce the consumption of unhealthy options.

Future research should explore interventions aimed at improving culinary skills and assessing their long-term impact on dietary habits and health outcomes. Particular attention should be given to underserved or vulnerable populations, where barriers such as time constraints, economic limitations, and access to fresh ingredients may hinder healthy culinary practices. Comparative studies across different cultural contexts could also provide a broader understanding of how culinary behaviors are shaped by societal norms and available resources. Additionally, the demographic trends observed, including the predominance of higher education among participants, suggest that interventions should be inclusive and sensitive to socioeconomic disparities. Finally, the broader applicability and long-term impact of such interventions should be further explored to ensure they are accessible and effective across different social groups and contexts.

Longitudinal studies evaluating the outcomes of culinary education initiatives could provide valuable insights into the sustained benefits of these programs on public health. This study design should be replicated in other countries and across diverse cultural and socioeconomic contexts to facilitate a comprehensive analysis and comparison of the study findings and to determine their generalizability. Such comparative studies would enhance our understanding of how culinary knowledge and practices influence dietary habits and health outcomes in different populations, ultimately informing public health strategies tailored to specific regional needs.

## Figures and Tables

**Figure 1 nutrients-17-01635-f001:**
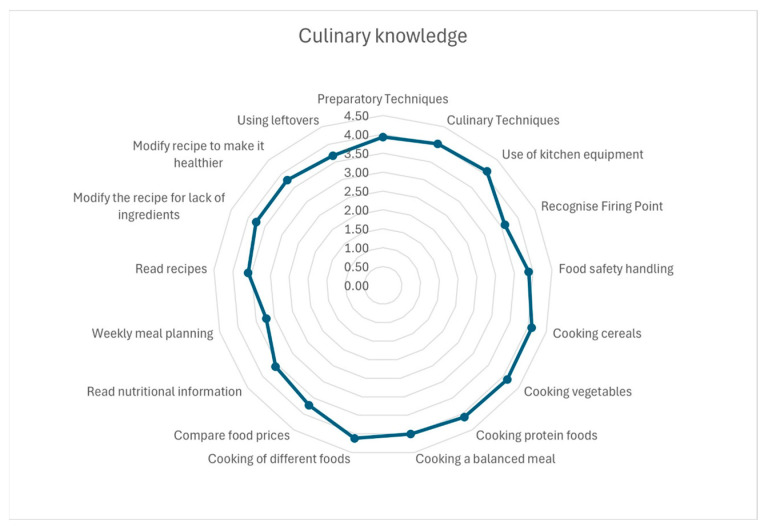
Participants’ confidence in various culinary techniques and activities.

**Figure 2 nutrients-17-01635-f002:**
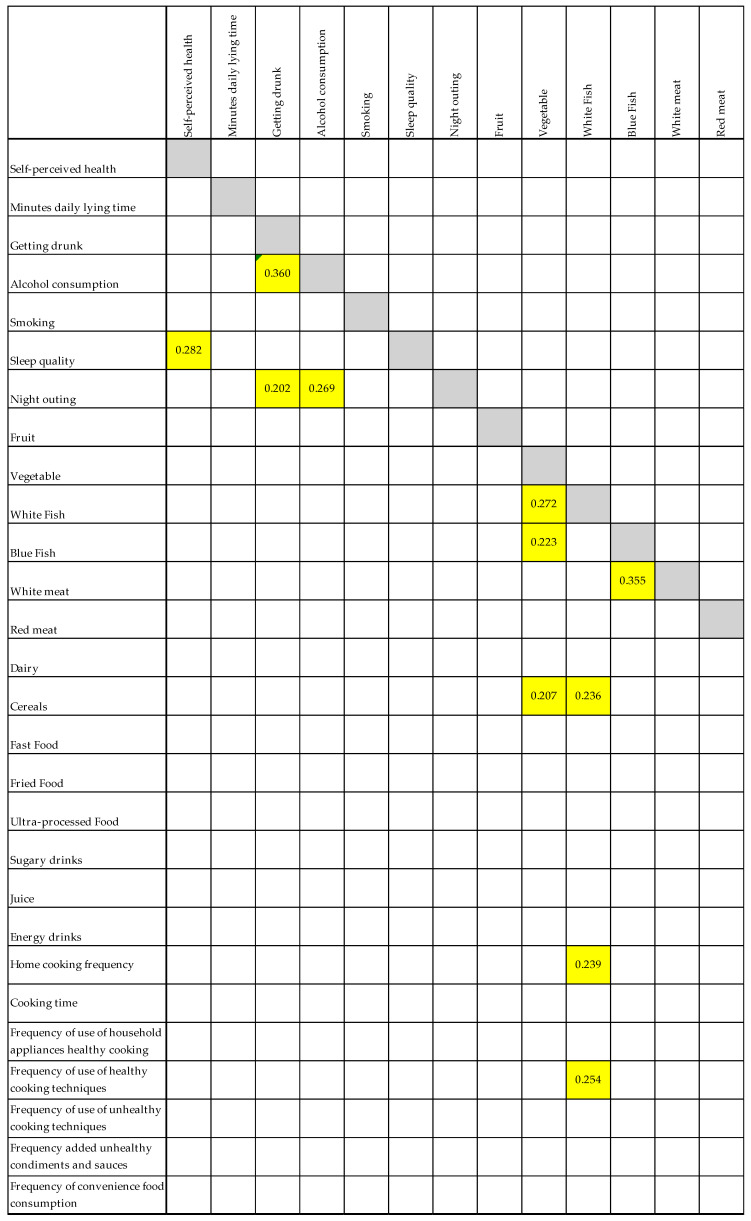
Spearman’s correlation for each pair of nutritional, lifestyle, and culinary habits variables. Legend: White cell = no correlation (0.00–0.19); yellow cell = low correlation (0.20–0.39); orange cell = moderate correlation (0.40–0.59).

**Figure 3 nutrients-17-01635-f003:**
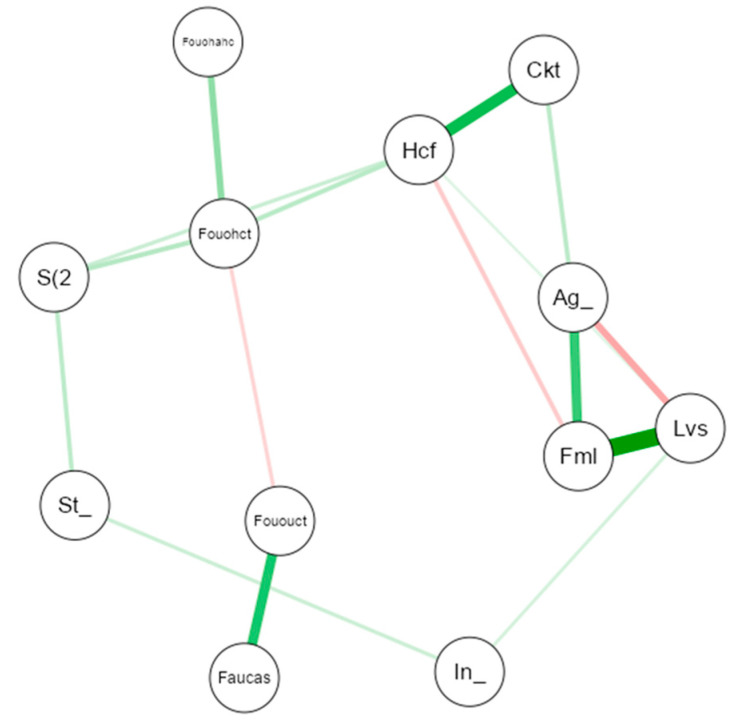
Gaussian network diagram between culinary habits and socio-demographic variables. (NOTE: The thickness of the lines reflects the magnitude of the relationship (partial correlation), while the colour indicates the direction: green for positive relationships and red for negative relationships).

**Table 1 nutrients-17-01635-t001:** Categorization of culinary habits variables.

Variables		Categorization
1. Home cooking weekly frequency		
	Usually, I do not cook	1
	1	2
	2–3	3
	4–5	4
	Every day or almost every day	5
2. Cooking daily time		
	Usually, I do not cook	1
	<1 h	2
	1–2 h	3
	2–3 h	4
	>3 h	5
3. Frequency of use of household appliances for healthy cooking		Likert scale 1–5 (1 = Never, 5 = Always)
4. Frequency of use of healthy cooking techniques		Likert scale 1–5 (1 = Never, 5 = Always)
5. Frequency of use of unhealthy cooking techniques		Likert scale 1–5 (1 = Never, 5 = Always)
6. Frequency of adding unhealthy condiments and sauces		Likert scale 1–5 (1 = Never, 5 = Always)
7. Channels for consulting recipes		Qualitative
8. Where participants learned how to cook		Qualitative
9. Frequency of convenience food consumption (every 3 days)		
	None	1
	1–3 times	2
	4–5 times	3
	>5 times	4
	I do not know	0

**Table 2 nutrients-17-01635-t002:** Socio-demographic characteristics of sample (N = 1534).

	N (%)
Sex	
Male	506 (33.0%)
Female	1028 (67.0%)
Age (years)	39.9 (15.0)
Male	38.6 (15.5)
Female	40.6 (14.7)
Age	
Young (18–30 years)	558 (36.4%)
Adults (31–65 years)	941 (61.3%)
Seniors (>65 years)	35 (2.3%)
Education level	
Basic	642 (41.9%)
Higher	892 (58.1%)
Income level	
Low	561 (36.6%)
Medium-high	797 (52.0%)
Do not know/no answer	176 (11.5%)
Living situation	
Alone	171 (11.1%)
With others	1363 (88.9%)
Family living	
Without relatives	323 (21.1%)
With relatives	1211 (78.9%)

**Table 3 nutrients-17-01635-t003:** Cooking habits (scale between 1 and 5 points).

Cooking Habits	Mean	(SD)
Home cooking frequency	4.08	(1.33)
Cooking time	2.44	(0.79)
Frequency of use of household appliances for healthy cooking	3.08	(1.30)
Frequency of use of healthy cooking techniques	3.84	(1.08)
Frequency of use of unhealthy cooking techniques	1.91	(0.80)
Frequency of adding unhealthy condiments and sauces	1.93	(0.90)
Frequency of convenience food consumption	1.56	(1.10)

**Table 4 nutrients-17-01635-t004:** Scores on culinary knowledge (scale between 1 and 5 points).

Culinary Knowledge	Mean	(SD)
Cooking vegetables	4.13	(1.13)
Cooking of different foods	4.12	(1.13)
Cooking cereals	4.10	(1.13)
Cooking protein foods	4.10	(1.10)
Use of kitchen equipment	4.09	(1.12)
Culinary techniques	4.02	(1.15)
Cooking a balanced meal	4.00	(1.11)
Preparatory techniques	3.93	(1.25)
Food safety handling	3.88	(1.14)
Modify recipe to make it healthier	3.77	(1.30)
Modify the recipe for lack of ingredients	3.76	(1.28)
Compare food prices	3.73	(1.22)
Using leftovers	3.68	(1.33)
Recognize firing point	3.61	(1.20)
Read recipes	3.59	(1.35)
Read nutritional information	3.57	(1.36)
Weekly meal planning	3.22	(1.39)
Total Culinary education	61.2	(17.0)

**Table 5 nutrients-17-01635-t005:** Health variables and nutritional and lifestyle habits (scale between 1 and 4 points, except for daily lying time, physical activity, and BMI).

Healthy Variables and Nutritional and Lifestyle Habits	Mean	(SD)
Minutes week moderate exercise	535	(15,492)
Minutes daily lying time	276	(2820)
Minutes week intense exercise	96.7	(208)
BMI	24.20	(4.26)
Self-perceived health	3.98	(0.75)
Vegetable	3.45	(0.75)
Dairy	3.35	(0.94)
Sleep quality	3.31	(0.85)
Body image	3.13	(1.26)
Obesophobia	3.01	(1.34)
Cereals	2.70	(1.01)
Getting up rested	2.60	(0.57)
White meat	2.54	(0.68)
Fast food	2.43	(0.76)
No control	2.41	(1.14)
Sleeping hours	2.41	(0.73)
Fruit	2.35	(0.75)
Fried food	2.32	(0.78)
Ultra-processed food	2.31	(0.91)
Legumes	2.00	(0.59)
Blue fish	1.84	(0.59)
Alcohol consumption	1.83	(0.87)
Coffee	1.76	(0.71)
Read meat	1.74	(0.69)
White fish	1.72	(0.57)
Sugary drinks	1.43	(0.52)
Night outing	1.34	(0.52)
Smoking	1.32	(0.73)
Juice	1.23	(0.54)
Getting drunk	1.16	(0.48)
Energy drinks	1.06	(0.30)

**Table 6 nutrients-17-01635-t006:** Health variables and nutritional and lifestyle habits compared to cooking habits.

	Mean (SD)	Mean (SD)	Mean (SD)		
Healthy Variables and Nutritional and Lifestyle Habits	Unhealthy Culinary Habits (UCH)	Need Improvement (NI)	Healthy Culinary Habits (HCH)	*p*-Value ^$^	
BMI	23.3 (4.85)	24.3 (4.19)	24.0 (4.34)	0.043	
Self-perceived health	3.87 (0.99)	3.89 (0.78)	4.07 (0.70)	<0.001 *	UCH vs. NI: 0.991; UCH vs. HCH: 0.816; NI vs. HCH: <0.001
Obesophobia	2.27 (1.33)	2.96 (1.35)	3.08 (0.70)	0.017	
No control	2.20 (1.01)	2.43 (1.19)	2.40 (1.08)	0.866	
Body image	2.80 (1.26)	3.08 (1.30)	3.19 (1.22)	0.171	
Minutes daily lying time	167 (207)	265 (1668)	291 (3704)	0.013 *	UCH vs. NI: 0.991; UCH vs. HCH: 0.871; NI vs. HCH: 0.009
Minutes weekly intense exercise	95.5 (119)	96.5 (195)	97.0 (222)	0.183	
Minutes weekly moderate exercise	123 (149)	938 (2159)	108 (263)	0.994	
Getting drunk	1.13 (0.35)	1.22 (0.56)	1.09 (0.37)	<0.001 *	UCH vs. NI: 0.937; UCH vs. HCH: 0.606; NI vs. HCH: <0.001
Alcohol consumption	2.13 (1.13)	1.90 (0.91)	1.74 (0.82)	0.003 *	UCH vs. NI: 0.761; UCH vs. HCH: 0.377; NI vs. HCH: 0.003
Smoking	1.27 (0.59)	1.41 (0.82)	1.22 (0.60)	<0.001 *	UCH vs. NI: 0.891; UCH vs. HCH: 0.800; NI vs. HCH: <0.001
Sleeping hours	2.47 (0.52)	2.37 (0.74)	2.46 (0.72)	0.098	
Sleep quality	3.47 (0.92)	3.24 (0.89)	3.39 (0.79)	0.003 *	UCH vs. NI: 0.460; UCH vs. HCH: 0.713; NI vs. HCH: 0.003
Getting up rested	2.73 (0.46)	2.57 (0.58)	2.63 (0.57)	0.077	
Night outing	1.47 (0.52)	1.39 (0.56)	1.29 (0.47)	<0.001 *	UCH vs. NI: 0.744; UCH vs. HCH: 0.263; NI vs. HCH: 0.001
Fruit	1.93 (0.80)	2.25 (0.75)	2.46 (0.73)	<0.001 *	UCH vs. NI: 0.281; UCH vs. HCH: 0.026; NI vs. HCH: <0.001
Vegetable	3.07 (0.70)	3.28 (0.83)	3.65 (0.60)	<0.001 *	UCH vs. NI: 0.319; UCH vs. HCH: <0.001; NI vs. HCH: <0.001
White fish	1.80 (0.41)	1.66 (0.55)	1.79 (0.58)	<0.001 *	UCH vs. NI: 0.488; UCH vs. HCH: 0.969; NI vs. HCH: <0.001
Blue fish	1.93 (0.59)	1.75 (0.57)	1.93 (0.60)	<0.001 *	UCH vs. NI: 0.451; UCH vs. HCH: 0.993; NI vs. HCH: <0.001
White meat	2.40 (0.63)	2.52 (0.68)	2.56 (0.69)	0.389	
Red meat	2.20 (0.68)	1.81 (0.71)	1.65 (0.67)	<0.001 *	UCH vs. NI: 0.081; UCH vs. HCH: 0.007; NI vs. HCH: <0.001
Dairy	3.40 (0.74)	3.27 (0.99)	3.44 (0.89)	0.002 *	UCH vs. NI: 0.989; UCH vs. HCH: 0.789; NI vs. HCH: 0.001
Cereals	2.47 (0.74)	2.58 (0.99)	2.84 (1.01)	<0.001 *	UCH vs. NI: 0.872; UCH vs. HCH: 0.221; NI vs. HCH: <0.001
Legumes	2.07 (0.46)	1.99 (0.60)	2.01 (0.59)	0.481	
Fast food	3.00 (0.93)	2.56 (0.77)	2.29 (0.72)	<0.001 *	UCH vs. NI: 0.087; UCH vs. HCH: 0.003; NI vs. HCH: <0.001
Fried food	3.40 (0.74)	2.52 (0.79)	2.08 (0.69)	<0.001 *	UCH vs. NI: <0.001; UCH vs. HCH: <0.001; NI vs. HCH: <0.001
Ultra-processed food	3.07 (0.96)	2.44 (0.93)	2.15 (0.85)	<0.001 *	UCH vs. NI: 0.051; UCH vs. HCH: 0.001; NI vs. HCH: <0.001
Sugary drinks	1.60 (0.83)	1.53 (0.70)	1.31 (0.59)	<0.001 *	UCH vs. NI: 0.965; UCH vs. HCH: 0.177; NI vs. HCH: <0.001
Juice	1.60 (0.99)	1.26 (0.55)	1.20 (0.51)	0.003 *	UCH vs. NI: 0.357; UCH vs. HCH: 0.094; NI vs. HCH: 0.008
Energy drinks	1.00 (0.00)	1.09 (0.36)	1.04 (0.22)	0.007 *	UCH vs. NI: 0.547; UCH vs. HCH: 0.747; NI vs. HCH: 0.007
Coffee	1.33 (0.49)	1.74 (0.69)	1.79 (0.72)	0.021	

^$^ Kruskal–Wallis test; * statistically significant.

**Table 7 nutrients-17-01635-t007:** Health variables and nutritional and lifestyle habits compared to culinary knowledge.

	Mean (SD)	Mean (SD)	Mean (SD)		
Healthy Variables and Nutritional and Lifestyle Habits	Poor Culinary Knowledge (PCK)	Average Culinary Knowledge (ACK)	Good Culinary Knowledge (GCK)	*p*-Value ^$^	
BMI	25.00 (3.86)	24.10 (4.26)	24.10 (4.30)	0.109	
Self-perceived health	3.98 (0.90)	3.90 (0.76)	4.02 (0.73)	0.009 *	PCK vs. ACK: 0.310; PCK vs. GCK: 0.989; ACK vs. GCK: 0.005
Obesophobia	2.92 (1.35)	2.96 (1.38)	3.05 (1.31)	0.337	
No control	2.42 (1.18)	2.46 (1.17)	2.39 (1.11)	0.496	
Body image	3.01 (1.28)	3.11 (1.31)	3.16 (1.23)	0.494	
Minutes daily lying time	185 (413)	244 (1576)	305 (3476)	0.537	
Minutes weekly intense exercise	90.5 (117)	104 (125)	92.9 (251)	0.400	
Minutes weekly moderate exercise	98.2 (125)	1211 (25,798)	164 (1550)	0.285	
Getting drunk	1.22 (0.65)	1.19 (0.52)	1.13 (0.44)	0.057	
Alcohol consumption	1.90 (1.01)	1.87 (0.88)	1.79 (0.85)	0.242	
Smoking	1.41 (0.84)	1.38 (0.79)	1.27 (0.67)	0.018 *	PCK vs. ACK: 0.981; PCK vs. GCK: 0.272; ACK vs. GCK: 0.020
Sleeping hours	2.25 (0.90)	2.36 (0.73)	2.46 (0.71)	0.009 *	PCK vs. ACK: 0.531; PCK vs. GCK: 0.007; ACK vs. GCK: 0.006
Sleep quality	3.08 (1.06)	3.27 (0.86)	3.36 (0.81)	0.010 *	PCK vs. ACK: 0.493; PCK vs. GCK: 0.009; ACK vs. GCK: 0.009
Getting up rested	2.57 (0.64)	2.55 (0.56)	2.64 (0.57)	0.025	
Night outing	1.30 (0.53)	1.40 (0.54)	1.31 (0.50)	0.011 *	PCK vs. ACK: 0.157; PCK vs. GCK: 0.449; ACK vs. GCK: 0.008
Fruit	2.33 (0.81)	2.23 (0.79)	2.43 (0.71)	<0.001 *	PCK vs. ACK: 0.582; PCK vs. GCK: 0.449; ACK vs. GCK: <0.001
Vegetable	3.20 (0.98)	3.25 (0.84)	3.60 (0.62)	<0.001 *	PCK vs. ACK: 0.968; PCK vs. GCK: <0.001; ACK vs. GCK: <0.001
White fish	1.75 (0.63)	1.67 (0.55)	1.75 (0.57)	0.036	
Blue fish	1.81 (0.64)	1.76 (0.61)	1.89 (0.57)	<0.001 *	PCK vs. ACK: 0.811; PCK vs. GCK: 0.386; ACK vs. GCK: <0.001
White meat	2.48 (0.64)	2.51 (0.67)	2.56 (0.70)	0.266	
Red meat	1.77 (0.66)	1.79 (0.70)	1.70 (0.69)	0.050	
Dairy	3.28 (1.03)	3.29 (0.94)	3.39 (0.93)	0.132	
Cereals	2.47 (0.97)	2.53 (1.02)	2.83 (0.98)	<0.001 *	PCK vs. ACK: 0.811; PCK vs. GCK: 0.002; ACK vs. GCK: <0.001
Legumes	1.94 (0.59)	1.96 (0.60)	2.03 (0.59)	0.047	
Fast food	2.36 (0.85)	2.53 (0.80)	2.38 (0.73)	0.001 *	PCK vs. ACK: 0.230; PCK vs. GCK: 0.991; ACK vs. GCK: 0.002
Fried food	2.41 (0.91)	2.48 (0.81)	2.21 (0.73)	<0.001 *	PCK vs. ACK: 0.847; PCK vs. GCK: 0.074; ACK vs. GCK: <0.001
Ultra-processed food	2.31 (0.91)	2.42 (0.93)	2.24 (0.88)	0.001 *	PCK vs. ACK: 0.583; PCK vs. GCK: 0.734; ACK vs. GCK: <0.001
Sugary drinks	1.57 (0.86)	1.47 (0.70)	1.38 (0.61)	0.011	
Juice	1.27 (0.54)	1.25 (0.58)	1.22 (0.52)	0.387	
Energy drinks	1.14 (0.46)	1.09 (0.38)	1.04 (0.22)	0.005 *	PCK vs. ACK: 0.472; PCK vs. GCK: 0.010; ACK vs. GCK: 0.022
Coffee	1.78 (0.81)	1.78 (0.72)	1.75 (0.69)	0.675	

^$^ Kruskal–Wallis test; * statistically significant.

## Data Availability

The data presented in this study are available upon reasonable request to the corresponding author due to privacy concerns.

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
