# Peer review of "Culinary Habits and Health: Analyzing the Impact of Cooking Practices and Knowledge Among Spanish Young Adults"

_nutrients, 2025, doi:10.3390/nu17101635_

Round 1
Reviewer 1 Report
Comments and Suggestions for Authors
Dear author,
The manuscript title is “Culinary Habits and Health: Analyzing the Impact of Cooking Practices and Knowledge Among Spanish Young Adults” and it aims to assess the relation between knowledge, cooking habits and lifestyle/health of young adults in Spain.
The topic falls within the aims and scope of the journal and it is important in terms of Public Health.
Some particular suggestions/comments will be done here:
- Lines 26/27 – Cooking frequency – higher or lower? – was moderately correlated?
- Line 28 – strong connections does not seem very scientific, can the authors please rephrase with concrete data?
- Lines 28/29 – again, maybe it would be better if the authors could clearly present the tendencies of the associations in concrete
- Line 35 – usually one does not repeat in keywords words that are already in the title
- Lines 73/74 – well, this is debatable but…
- Line 82 – this reviewer missed in background a little bit more about the importance of these social, cultural and even emotional background in eating habits.
- Line 114 – are the XXX ok?
- Line 123 – the presentation of the instruments seems quite confusion to this reviewer; plus, the questionnaires applied must be shown in supplementary material; this reviewer needs to see the questions and options given, it is the most important part of the methodology, it is the tool used, it should be presented at submission
- Lines 148/149 – this may be a problem as the authors should write gender instead of sex, and obviously include non-binary otherwise it is not for sure representative of the Spanish population
- Line 149 – age – this reviewer though this was a questionnaire only for young adults…it is what is suggested in the title, abstract…
- Line 154 – What was the rational to the classification of the municipalities? Because this reviewer does not believe there must be huge differences between a village with 2,000 or 10,000 inhabitants
- Line 159 – Sorry but this reviewer can not understand how culinary habits were assessed teaching respondents how to cook!
- Table 1 – the number of the variables is not correct
- Lines 176/177 – the reviewers must see the questionnaire applied
- Lines 179 – 182; 186 - 188 – how did the authors validate this scale?
- Lines 236-238 - what biases do you predict for this minority of men and for this age range?
- Line 238 – young (typo error)
- Table 2 – Mean (SD)???
- Line 246 and elsewhere – Authors must be careful in the way of writing; as this was not an observational study, one can not affirm that the respondents “use” but only that “the respondents self-report that use”
- Tables 3 and 4 and elsewhere – maybe it is better to be said in the title what is the scale range to make the result clearer?
- Tables 4/5 and elsewhere – This reviewer suggest the authors to write the variables in an ascending or descending order
- Tables 5/6/7 and elsewhere – health variables and not healthy I suppose
- Figure 2 – it is not possible to be read
- Lines 405 – 408 – that is why these results must be discussed with “tweezers”
- Line 455 – 61.3% of your sample is not from young adults (18-30) but from adults (31 – 65). This reviewer thinks you can not call “young” to someone who is 65 or even if the mean age is 40, sorry, 40 is not young anymore. Unless the authors have a rationale to explain this.
- Discussion: in the discussion nothing is said about the lifestyles that were assessed and the alcohol and tobacco consumption
Author Response
Dear Reviewer
Thank you very much for your thorough review and valuable suggestions. We truly appreciate the time and effort you dedicated to improving our manuscript. Your insightful comments have helped us to strengthen the quality and clarity of our work. Please find attached a document with detailed responses to each of your comments.
Best regards
Dr. Elena Sandri

Reviewer 2 Report
Comments and Suggestions for Authors
This study aimed to examine the culinary knowledge and skills of young Spanish adults and their relationship to nutritional and lifestyle habits. As the results, participants with healthy culinary habits reported higher self-perceived health, and greater fruit and vegetable consumption, while those with unhealthy habits consumed more fast food. Furthermore, higher culinary knowledge was linked to better dietary choices, more vegetables and cereals, and lower smoking rates. The authors examined the association of culinary practices and knowledge with the lifestyle habits of Spanish young adults, and the reviewer believes that the results of this study may provide useful information in terms of preventing the development of future obesity, diabetes, and cardiovascular disease. Furthermore, the reviewer thinks that this study had a large number of participants and was appropriately analyzed. However, there are several limitations in this study.
- The participants in this study included 33.0% men and 67.0% women, resulting in data biased towards women. Although it cannot be denied that women have more opportunities to cook, the reviewer believes that one of the limitations of this study is that the results were biased toward a smaller number of men and a larger number of women.
- In relation to the above, is it assumed that participants in this study will cook their own meals? It is expected that in some cases the participant will not cook, but rather a person living with them will do so, and in other cases the participant will cook themselves. The reviewer thinks that culinary practices and knowledge differ between participants who cook for themselves and those who do not.
- In the “Potential eating disorders variables”, the authors asked participants whether they had been diagnosed with an eating disorder. However, the exclusion criteria for participants in this study did not include those with eating disorders. How did the authors treat participants with eating disorders in this study?
- Exclusion criteria for this study included individuals with chronic diseases that affect dietary habits. Please explain any specific chronic diseases.
- There are many items in the tables in this article that do not have units. Please add units to any items that do not have units.
- Comments: In this study, the authors examine the association of culinary knowledge and skills of young Spanish adults with their lifestyle habits. The reviewer would be very interested to know how much total energy and nutrient intake the participants in this study were receiving. If the authors have data on total energy and macronutrient intakes, we would appreciate them providing these.
Author Response

(The authors gave the same response as above.)

Reviewer 3 Report
Comments and Suggestions for Authors
The manuscript is interesting and well prepared. I have a few small comments: (1) Figure 2 is illegible - it is necessary to improve the graphic quality of this figure; (2) the references are not formatted according to MDPI guidelines - correct this.
Author Response

(The authors gave the same response as above.)

Round 2
Reviewer 1 Report
Comments and Suggestions for Authors
Dear authors,
Thank you for your revised manuscript, this reviewer believes that good improvements were made.
Only few things remain
- The authors kept in the title “young”, I must remember that “– 3% of your sample is not from young adults (18-30) but from adults (31 – 65). This reviewer thinks you can not call “young” to someone who is 65 or even if the mean age is 40, sorry, 40 is not young anymore.”
- When said that maybe it would be better if the authors could clearly present the tendencies of the associations in concrete, this reviewer meant that in this phrase:
“The network analysis showed that the frequency of cooking at home is positively associated with gender and age. A similar association is found for time spent cooking. Positive associations were also found between living with the family and time spent cooking, while age showed a negative correlation with living situation. “…It would be better to explicit clearly how the frequency is associated with gender/age. Women cook more frequently than men? Younger cook less frequently than older?
- This reviewer believes that the questionnaire must be added as supplementary material, of course, it is the main tool of methodology, but the Editor will decide best; it is not easy to evaluate as presented and it does not include all the questions made
Author Response
Dear reviewer
Thank you again for your time and attention in reviewing our manuscript and for the valuable advice received.
In the attached file you will find the response to your comments.
Kind regards
Dr. Elena Sandri

Reviewer 2 Report
Comments and Suggestions for Authors
I think all responses to reviewers' comments have been addressed satisfactorily.
I have no comments on the revised manuscript.
Author Response
Dear Reviewer
We thank you again for your time and attention in reviewing our manuscript and for the precious advice received.
We are very pleased that the revisions were adequate.
Kind regards
Dr. Elena Sandri